# InfoNet: Missing Information Retrieval in Multi-Stream Sensing Systems

## Abstract

Faulty sensors in a multiple input stream setup are more prone to corrupted input data streams, hindering the performance of Deep Neural Networks (DNN), which focus on deducing information from data. However, the relevant information among multiple input streams has correlations and contains mutual information. This paper utilizes this opportunity to retrieve perturbed information caused by corrupted input streams. We propose **InfoNet**, which estimates the information entropy at every element of the input feature to the network and retrieves the missing information in the input feature matrix. Finally, using the estimated information entropy and retrieved data, we introduce a novel guided replacement procedure to recover the complete information that is the input to the downstream DNN task. We evaluate the proposed algorithm for sound localization where audio streams from the microphone array are corrupted. We have recovered the performance drop due to the corrupted input stream and reduced the localization error with non-corrupted input streams. Finally, we assess the potential of using the proposed algorithm for retrieving information in other sensing modalities, e.g., wireless signal-based source localization.

## 1 Introduction

Deep learning has shown promising results in extracting meaningful information from sensor data in the last few years. Currently, deep learning is largely used in various domains including event detection Alitaleshi et al. (2023), multi-camera multi-object tracking Wang et al. (2023), speaker and vehicle localization Bohlender et al. (2023); de Godoy et al. (2018), speaker separation Nugraha et al. (2016), cybersecurity Jiang et al. (2018), health monitoring Yuan et al. (2018), and emotion recognition Chao et al. (2018). With the popularity of multi-sensor devices, e.g., microphone arrays in Amazon Echo , Google Home , and wireless earbuds , and camera array in recent smartphones, the probability of having faulty sensors has increased. These faulty sensors result in missing or corrupted information in one or more input streams due to uncertainty, e.g., unstable communication medium Hyadi et al. (2016), stochastic energy Islam & Nirjon (2020); Monjur et al. (2023), and sensor failure Li et al. (2020a). However, traditional a deep neural networks (DNNs) are not resilient to these faulty data Li et al. (2020b) and perform poorly with missing or corrupted portions in inputs Roy et al. (2020); Yin & Hou (2016).

Though statistical methods Jerez et al. (2010) like mean imputation Allison (2001), hot deck Little & Rubin (2019), and multiple imputations Rubin (1996) can recover missing data portions, these imputation techniques fail whenever the missing data sequence is long Hasan et al. (2021). Besides, the statistical feature is insufficient for higher dimensional and complex data distributions in real-world applications Al-Janabi & Alkaim (2020), including image processing, acoustic sensing, and Radio Frequency (RF) imaging. Literature shows extracted features, e.g., generalized cross-correlation and spectral power, perform significantly better than raw time-domain as input to a DNN when a large training corpus is absent Rajanna et al. (2015). Researchers have developed ultra-low power devices for feature extraction from different sensors for various tasks with techniques like analog signal processing and time-mode analog filter banks Yang et al. (2019); Ray & Kinget (2023).

Therefore, there is a need for more sophisticated techniques for recovering or proxying these missing or corrupted complex features to avoid erroneous DNN inference output.

Recent works on mutual information and conditional entropy can quantify the relationship between the variables Vergara & Estévez (2014). Though these methods can be exploited to identify the corrupted portion in the complex feature set, they are insufficient to recover the information for maintaining the performance of the DNN-based sound source localization. On the other hand, recent works on information estimationHjelm et al. (2018); Poole et al. (2019) present the opportunity to recover the corrupted data. However, blindly recovering all data without knowing which one is corrupted results in the alteration of the correct data, which results in poor performance in the DNN.

This paper proposes **InfoNet**, a *generalized* algorithm that retrieves the information from a corrupted feature set to recover the inference performance loss due to corrupted input data streams.*It combines information entropy estimation and conditional interpolation to identify the corrupted portions in the feature set and estimates the true feature values.* **InfoNet** consists of three novel components – (1) **early attention** (EA) that estimates the information entropy at each element of the calculated features to understand the effect of the corrupted data stream on that element; (2) **deep conditional interpolator** (DCI) that estimates missing information utilizing the estimation relation; and (3) **guided replacement** (GR) that takes the information entropy from the EA and replaces the affected element from the initially generated features with the interpolated features from DCI without impacting the element unaffected by corrupted data. This newly replaced feature serves as the input to the downstream DNN task, e.g., sound source localization, for instance.

For evaluation, we focus on sound source localization with a microphone array, which is a challenging component of various applications, including augmented and virtual reality Ahrens et al. (2019); Argentieri et al. (2015) and robotics Argentieri et al. (2015). Here we consider each microphone of a microphone array as the input stream of a multi-stream sensor system. We first use the popular **DCASE dataset** Politis et al. (2021b) (*Dataset 1*) and achieve *36.91 ± 3.56%* less degree of arrival (DoA) error with the corrupted data stream. Next, to mitigate the lack of a large sound source localization dataset, we simulated and published a **large audio dataset** (*Dataset 2*) of 50 hours with 10 different environments. We have reduced the DoA on the corrupted dataset by $22.54 \pm 18.30\%$ using **InfoNet**.

Finally, we show that the proposed algorithm is suitable for other application domains, such as wireless localization, to prove the generalizability of **InfoNet**. We evaluate on *wireless source localization* task using **WiFi channel state information (CSI) dataset** Ayyalasomayajula et al. (2020) with *75% missing information* and achieved 25.52% improvement in $90^{th}$ percentile localization error.

## 2 PROBLEM FORMULATION AND MATHEMATICAL FOUNDATION

Let us consider $X$ as the raw data dependent on a set of $n$ random variables, $\mathcal{S} = \{x_1, x_2, ..., x_n\}$. Then $X$ can be represented by $X = f(\mathcal{S})$. $X$, is then mapped to the feature domain data, $F$, by a feature extraction function, $G$. Thus, $F = G(X)$.

Though $F$ is considered as a full rank matrix in traditional deep learning, corruption in sensor data results in one or some missing or corrupted element in set $\mathcal{S}$. As a result, full-rank data, $X$, becomes low-rank, $\tilde{X}$ producing a rank-deficient $\tilde{F}$. In this paper, *we focus on recovering $\hat{F}$ from $\tilde{F}$, where $\hat{F} \approx F$.*

Since $F$ is a function of $X$, and $X$ depends on a set of random variables ($\mathcal{S}$), the amount of information content or entropy, i, at every element of $F$ depends on the same set of random variables, $S$. Thus the information entropy at the $k^{th}$ element of $F$, $i_k$, is formulated as

$$i_k = -G(f(\mathcal{S}_k)) \log G(f(\mathcal{S}_k)) \tag{1}$$

Here, $i_k$ is a real number, and $i_k \in [0, 1]$. $i_k = 1$ represents the presence of all information, while $i_k = 0$ indicates a complete absence of information. Any value between 0 and 1 implies *fractional missing information* and quantifies the amount of missing information in the feature map $\tilde{F} = G(\tilde{X})$.

Orchard et al. introduced the computation principle of fractional missing information Orchard & Woodbury (1972), which states that the likelihood of complete data, $L(\theta|X)$, can be factored into the likelihood of the observed lower-ranked data, $L(\theta|\tilde{X})$, and the density of rank-deficient data given the observed data, $f(Z|\theta, \tilde{X})$, where $Z$ and $\theta$ are missing data and model parameter, respectively.

$$L(\theta|X) = L(\theta|\tilde{X})f(Z|\theta, \tilde{X}) \tag{2}$$

The first derivative of the complete data log-likelihood can be represented as

$$\frac{\partial logL(\theta|X)}{\partial \theta} = \frac{\partial logL(\theta|\tilde{X})}{\partial \theta} + \frac{\partial logf(Z|\theta, \tilde{X})}{\partial \theta} \tag{3}$$

The covariance matrix of equation 3 defines the information matrix, $\hat{F}_X$, which is derived by the following equation–

$$\hat{F}_X = Cov\left(\frac{\partial logL(\theta|X)}{\partial \theta}\right); \quad \hat{F}_{\tilde{X}} = Cov\left(\frac{\partial logL(\theta|\tilde{X})}{\partial \theta}\right); \quad \hat{F}_{X|\tilde{X}} = Cov\left(\frac{\partial logf(Z|\theta, \tilde{X})}{\partial \theta}\right)$$

Here, $Cov(\cdot)$ is the covariance. Finally, equation 3 can be written in terms of the complete information $J_X$ which is the sum of available low-rank information $J_{\tilde{X}}$ and $J_{X|\tilde{X}}$ missing information matrix.

$$\hat{F}_X = \hat{F}_{\tilde{X}} + \hat{F}_{X|\tilde{X}} \tag{4}$$

The rank-deficient $\tilde{F}$ consists of information content and noise content. The latter arises due to the missing elements of $\mathcal{S}$. Hence, the information content, $\hat{F}_{\tilde{X}}$, can be written in terms of $\tilde{F}$ as equation 5, here $I$ is the element-wise information entropy of $\tilde{F}$. So, the missing information matrix $\hat{F}_{X|\tilde{X}}$ can be obtained by estimating the values of the missing elements in noise contents from the existing information contents. If $\bar{\bar{F}}$ is the estimated missing information then $\hat{F}_{X|\tilde{X}}$ is presented by equation 6.

$$\hat{F}_{\tilde{X}} = I \otimes \tilde{F} \tag{5} \qquad\qquad \hat{F}_{X|\tilde{X}} = (1 - I) \otimes \bar{\bar{F}} \tag{6}$$

With equation 5 and 6, equation 4 can be re-written as

$$\hat{F} = \hat{F}_X = I \otimes \tilde{F} + (1 - I) \otimes \bar{\bar{F}} \tag{7}$$

## 3 PROPOSED METHODOLOGY

**InfoNet** maps the low-ranked feature matrix, $\tilde{F}$, of Equation 7 into the full-ranked feature matrix, $\hat{F} \approx F$ by estimating the missing information matrix for a given task using a three-step approach. Figure 1 shows the overall architecture of **InfoNet**, which sits between the feature extraction module and the application-dependent *downstream task*. **InfoNet** consists of three main components – (1) early attention (EA) that estimates the low-rank information and measures the element-wise information entropy, $I$ in $\tilde{F}$ (Equation 5); (2) deep conditional interpolator (DCI) which interpolates the missing elements ($\bar{\bar{F}}$ in Equation 6) from the available low-rank feature matrix, $\tilde{F}$; and (3) guided replacement (GR) which combines the less informative elements of the original feature set with the interpolated feature values in the guidance of element-wise information entropy. This combining process is governed by the equation 7. The resultant enhanced feature set, $\hat{F}$ is fed to the downstream task for inference.

### 3.1 EARLY ATTENTION (EA)

This step aims to estimate the information entropy, $I$, for each element of the low-rank feature set $\tilde{F}$. To achieve this, we design an early attention module that estimates the channel and spatial information entropy Park et al. (2018). Low-rank feature set $\tilde{F} \in \mathbb{R}^{d_{\tilde{X}}}$ is the input to the EA, where $d_{\tilde{X}}$ is the dimension of $\tilde{F}$. The left-hand side of Figure 1 shows the details of the proposed early attention architecture which has two main components.

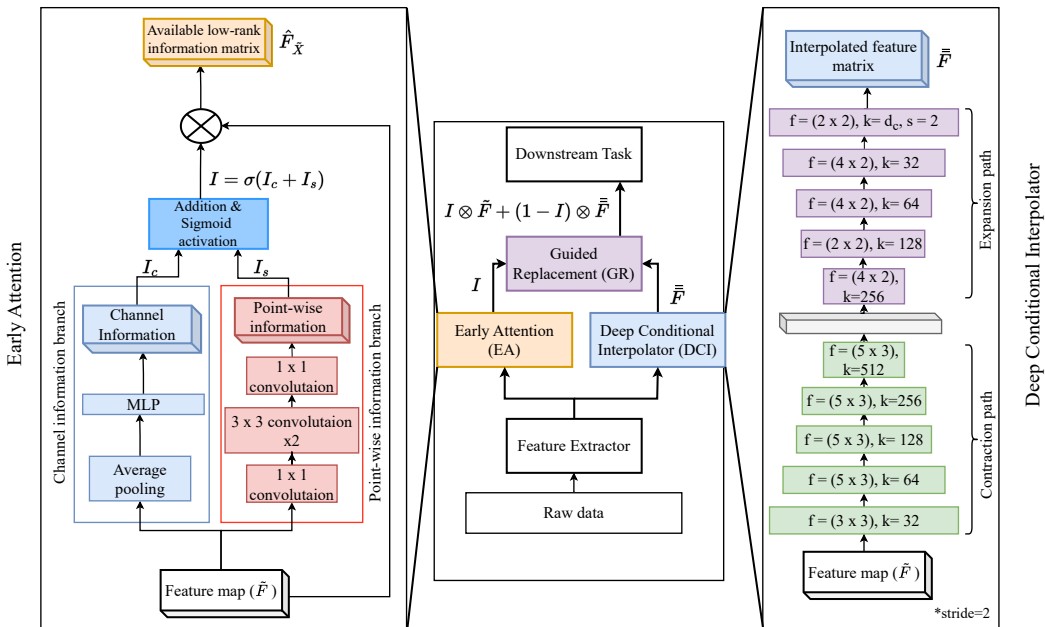

Figure 1: Architecture of proposed **InfoNet**. Note that only the highlighted components are the contributions of this paper.

The first component of EA is the *channel information branch*, $f_\gamma(\tilde{F})$, parameterized with $\gamma$. This branch performs average pooling, followed by a multi-layer perceptron on the input to extract the channel information entropy. Average pooling feeds gradients across all indices and thus enables the model to learn more robust features while more accurately depicting the overall strength of a feature. At this point, we expand the feature by an expansion factor, $e$. The multi-layer perceptron consists of three fully connected layers with batch normalization and rectified linear unit (ReLU) after the first two layers. Finally, this branch outputs a vector $I_c$ that measures of global information on each channel.

The second component of EA is the *point-wise information branch*, $f_\lambda(\tilde{F})$, parameterized with $\lambda$. The input, $\tilde{F}$, first passes through a convolution layer with a kernel size of $(1 \times 1)$ and a channel reduction factor of $r$. Then it goes through $n$ number of dilated convolution layers with a kernel of $(3 \times 3)$ with dilation value $d$, batch normalization, and ReLU activation. In our architecture, n = 3 and d = 2. The dilated convolution increases the respective field and enables us to leverage contextual information. Finally, this contextual information passes through another convolutional layer of kernel size $(1 \times 1)$ and outputs $I_s$, which estimates the point-wise information of the input feature.

$$I_c = f_\gamma(\tilde{F}) \qquad (8) \qquad\qquad I_s = f_\lambda(\tilde{F}) \qquad (9)$$

Next, we broadcast $I_s$ to $\mathbb{R}^{d_{\tilde{x}}}$ to get information entropy estimation of the same dimension as $\tilde{F}$ and perform element-wise addition on $I_c$ and $I_s$. Element-wise addition ensures smooth gradient flow He et al. (2016). Finally, the sigmoid function maps the results between 0 and 1, where a value closer to 1 indicates higher information availability and vice versa. Therefore, Equation 5 can be rewritten as –

$$\hat{F}_{\tilde{X}} = \sigma(I_c + I_s) \otimes \tilde{F} \qquad (10)$$

Here, $\otimes$ denotes element-wise multiplication, $\sigma(I_c + I_s)$ is the information entropy, $I$ (equation 5), and $\sigma$ is sigmoid function.

## 3.2 Deep Conditional Interpolator (DCI)

The deep conditional interpolator (DCI) estimates the missing information by exploiting relations among the available channels and is parameterized by $\beta$, $\bar{\bar{F}} = f_\beta(\tilde{F})$, which is used in Equation 6. The right-hand side of Figure 1 shows the architecture of DCI. It takes $\tilde{F}$ as input and provides interpolated feature $\bar{\bar{F}}$ as output. Thus,

$\bar{\bar{F}} = f_\phi(x_j | x_1, x_2, ..., x_{j-1}, x_{j+1}, ..., x_n)$. Here, $x_j$ is the corrupted or missing data stream. We propose a function $f_\phi$ that takes the low-rank feature and interpolates it to reach its full-rank. We choose an auto-encoder for estimating $f_\phi$ as it can learn compressed intermediate representation. They efficiently interpolate by semantically mix characteristics from data Berthelot et al. (2018) and are used in qualitative experimentation and latent variable generation Dumoulin et al. (2016); Ha & Eck (2017).

DCI contains two paths – (1) contraction and (2) expansion. The contraction path has $n$ consecutive downsampling blocks, where each block consists of a convolution layer with ReLU and batch normalization. In our implementation, $n = 5$, kernel size $= 5 \times 3$, and ReLU slope $= 0.2$. Though we downsample the feature using stride $= 2$ on each step, we double the number of output feature channels. The output of the contraction path is $F_{con} \in \mathbb{R}^{d_c}$, where $F_{con}$ stands for the latent intermediate representation of the input feature set, $\tilde{F}$, and $d_c$ represents the intermediate feature dimension.

The expansion path passes the input $F_{con}$ through 5 consecutive upsampling blocks consisting of transposed convolution (kernel $= 4 \times 2$ or $2 \times 2$ and stride $= 2$) with batch normalization and leaky-ReLU activation. The last convolution uses a sigmoid activation with no batch normalization. After each up-sampling block, we half the number of output feature filters and set the last layer's output filter number to match the shape of the input feature map. The expansion path returns the interpolated feature matrix, $\bar{\bar{F}} = f_\beta(\tilde{F})$, where $\bar{\bar{F}} \in \mathbb{R}^{d_z}$.

### 3.3 GUIDED REPLACEMENT (GR)

We replace the low-rank $\tilde{F}$ with the interpolated feature $\bar{\bar{F}}$. The whole replacement process is guided by $\sigma(I_c + I_s)$. The $k^{th}$ element $\sigma(I_c + I_s)$ provides the amount of available information present at the $k^{th}$ element of $\tilde{F}$. Hence, $F_{X|\tilde{X}}$ in equation 6 can be quantified as follows

$$\hat{F}_{X|\tilde{X}} = (1 - \sigma(I_c + I_s)) \otimes \bar{\bar{F}} \tag{11}$$

Guided replacement utilizes the low-rank information matrix, $\hat{F}_{\tilde{X}}$, and missing information matrix, $\hat{F}_{X|\tilde{X}}$, to find the estimated full-rank enhanced feature matrix $\hat{F}_X$ from equation 7.

### 3.4 TRAINING

We train **InfoNet** jointly with the downstream task. Let us assume that the loss function of the downstream task is $\mathcal{L}_{DT}$. The impact of this loss, $\mathcal{L}_{DT}$, backpropagates through the downstream task as well as the channel and point-wise information branches ($f_\gamma$ and $f_\lambda$) of EA. However, DCI ($f_\beta$) has a separate training loss, $\mathcal{L}_{DCI}$, the squared $L_2$ distance between $F$ and $\bar{\bar{F}}$. $\mathcal{L}_{DCI}$ is defined by $\mathcal{L}_{DCI} = ||F - \bar{\bar{F}}||^2$. Note that we use two different optimizers to train the EA and downstream task with $\mathcal{L}_{DT}$ and DCI with $\mathcal{L}_{DCI}$ separately.

## 4 EXPERIMENTAL SETUP

This section describes the dataset, feature extraction, downstream task, baseline, data preparation, and evaluation metrics to evaluate **InfoNet** for sound source localization. Our implementation can be found here [1].

**Dataset.** We evaluate **InfoNet** on 2 multichannel sound source localization datasets.

*Dataset 1 – DCASE.* We use the DCASE2021 Task 4 dataset Politis et al. (2021a) which aims to localize sound sources regarding the degree of arrival (DoA). The dataset consists of 600 4-channel 1-minute spatial recordings of overlapping sound events sampled at $24KHz$. The development and evaluation contain 400 and 200 samples, respectively. A group of spatial room impulse responses (SRIR) determines the synthesis of these spatial recordings, and a maximum of 4 polyphony can be present in each recording. Moreover, multi-channel

---

[1]`https://github.com/hidethyself/InfoNet`

ambient noise is present where the noise level is scaled in signal-to-noise-ratio (SNR) and is randomly taken from $6dB$ to $30dB$.

*Dataset 2 – LargeSet.* To address the lack of a large dataset for sound source localization, we simulated and published a 50 **hours of multi-channel audio dataset** for the localization task. This dataset has 10 different environments and each audio contains a 1 second audio sampled at $44.1KHz$. We have used the UrbanSound8K dataset Salamon et al. (2014) as the sound source. The complete dataset simulation process and environment configurations are in the appendix A. The dataset can be found here Authors (2023).

**Feature Extraction.** We follow the feature extraction procedure provided by the DCASE challenge. We first perform 1024-point FFT with a 40 ms window length and 20 ms hop length at 24kHz and 44.1kHz on the raw time-domain audio data. Next, we calculate two features from the FFT – (1) multi-channel log-Mel-spectrogram in 64 Mel bands, and (2) generalized cross-correlation (GCC), where we truncate the GCC sequences to the same value as the Mel bins. Finally, we stack these two features to obtain the feature matrix, $F$.

**Corrupted Data Preparation.** To prepare a dataset with corrupted information, we randomly select a channel and replace the original audio with Gaussian noise of 0 mean and 1 standard deviation. We assume up to 50% simultaneous faulty channels and the number of missing channels belongs to the set $[0, 1, 2, 3]$. A $p\%$ missing data percentage (**MDP**) denotes that $p\%$ of the total audio length is corrupted with noise. Figure 2 shows an example raw audio where MDP is 100%, and only 1 channel is missing channel at any time.

**Downstream Task.** For the downstream task, we implement the SELDNet Adavanne et al. (2018) algorithms for sound event localization, which uses multi-channel audio to benefit from the spatial relationship among microphones. We consider each mic input as a random variable, $x_i$, and an element of set $\mathcal{S}$. Thus, any corrupted element in $\mathcal{S}$ results in a low-rank feature matrix $\tilde{F}$. The model comprises a convolutional recurrent neural network (CRNN) Cao et al. (2019) and a single full-connected (FC) layer. Detailed architecture of the CRNN is provided in appendix C.1 .

**Evaluation Baseline.** We evaluate the performance of **InfoNet**; we compare against the following baselines.

*Retrained with Missing Information.* We retrain the downstream task (***RTrain**$_p$*) for the SEL task, where $p$ represents *MDP* for the training and testing set.

*Time-Domain Recovery Techniques.* To evaluate the model performance against the time domain feature recovery techniques, we replace the Gaussian noises of the missing channel with the data from the most correlated channel (***REP**$_{corr}$*) and the average of all available channels (***REP**$_{avg}$*). Then, we pass the calculated features from these raw channels through the pre-trained downstream task.

**Evaluation Metrics.** We evaluate the performance of our sound event localization with three metrics – (1) *Degree of Arrival Estimation Error ($E_{DoA}$)* of the entire dataset, (2) *Event Localization F1-score when $E_{DoA} \leq 20^O$ ($F1_{20}$)*, and (3) *Localization Recall ($R_L$)* on each frame. The detailed definitions and derivations are provided in the appendix B.

## 5 Results

This section provides the results of our experiments by first comparing **InfoNet** with the baseline and showing how the performance varies with the proportion of missing information. Next, we perform an ablation study to show the effect of different components of **InfoNet** and some design choices.

### 5.1 Performance Analysis Compared to Baseline Algorithms

**DCASE.** Figure 3 compares $E_{DoA}$ among the baselines and **InfoNet**, where 75% of data are missing during inference. As the extracted feature loses spatial information in the presence of corrupted data, SELDNet's performance drops. We notice 58.16% of performance drop of SELDNet with 75% of MDP.

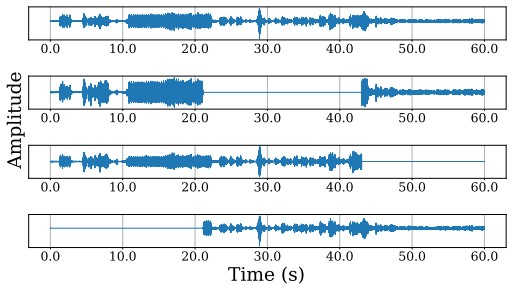

Figure 2: An example of raw audio with 100% missing data.

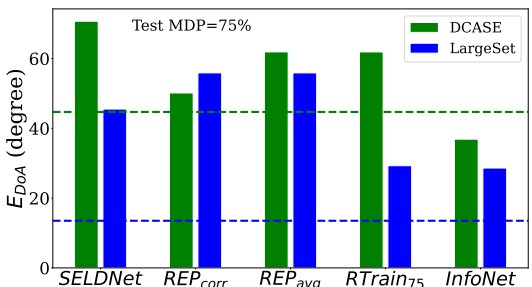

Figure 3: Comparison among different algorithms. Here, dotted lines represent the SELDNet's $E_{DoA}$ with uncorrupted data.

First, we evaluate the performance of two time-domain recovery techniques, which replace corrupted time-domain data and retrain the downstream task with the recovered data. Both $REP_{corr}$ and $REP_{avg}$, can recover 20.56 and 8.79 degrees, consecutively, from the $E_{DoA}$ drop by the baseline *SELDNet*. While these approaches fail to recover phase information, **InfoNet** considers channel and spatial relations to retrieve the information. Thus, **InfoNet** recovers 26.41% and 40.39% more performance than $REP_{corr}$ and $REP_{avg}$, respectively. Besides, **InfoNet** does not require prior knowledge of which mic is corrupted, which is essential for time-domain replacement models. Additionally, to increase efficiency, feature extraction is often performed in analog circuits de Godoy et al. (2018); Trevisi et al. (2015), making feature recovery more desirable than time-domain information recovery.

Next, we compare with $RTrain_p$, which is retrained with a corrupted feature set and recovers $E_{DoA}$ by 12.45% compared to the *SELDNet* which is trained with the uncorrupted feature set. On the contrary, **InfoNet** recovers 47.81% $E_{DoA}$.

**LargeSet.** In Figure 3, the performance of *SELDNet* drops by 70.32% compared to uncorrupted data. $REP_{corr}$ and $REP_{avg}$ drops $E_{DoA}$ by 75.83% and 75.61% than *SELDNet*, correspondingly. These time-domain recovery methods fail to improve the results as they can not incorporate the phase information. $RTrain_p$ recovers performance by 35.67% from *SELDNet*. **InfoNet** achieves 37.77% performance gain compared to SELDNet inference results in the presence of corrupted data. More results, including visual examples of GR, are added in the appendix D.

## 5.2 EFFECT OF MISSING DATA PERCENTAGE (MDP)

Table 1: Performance of SELDNet, $RTrain_p$ and **InfoNet** trained with $p$% MDP.

| MDP | | 0 | | | 25 | | | 50 | | | 75 | | | 100 | | |
|---|---|---|---|---|---|---|---|---|---|---|---|---|---|---|---|---|
| Dataset | Model | $F1_{20}$ | $E_{DoA}$ | $R_L$ | $F1_{20}$ | $E_{DoA}$ | $R_L$ | $F1_{20}$ | $E_{DoA}$ | $R_L$ | $F1_{20}$ | $E_{DoA}$ | $R_L$ | $F1_{20}$ | $E_{DoA}$ | $R_L$ |
| | SELDNet | 16.90 | 42.10 | 34.70 | 8.20 | 55.20 | 4.70 | 4.70 | 68.73 | 15.30 | 1.70 | 70.70 | 11.70 | 1.50 | 92.60 | 13.50 |
| DCASE | $RTrain_p$ | 16.90 | 42.10 | 34.70 | **14.70** | 59.10 | 30.70 | 10.30 | 63.30 | 26.50 | 11.40 | 61.90 | 26.10 | 9.10 | 67.20 | 25.90 |
| | InfoNet | **18.10** | **32.50** | **39.80** | 13.90 | **35.90** | **34.60** | **13.40** | **43.00** | **33.80** | **11.80** | **41.30** | **33.30** | **12.80** | **41.10** | **32.60** |
| | SELDNet | 85.94 | 16.41 | 75.35 | 85.86 | 16.43 | 75.23 | 72.58 | 25.81 | 56.96 | 42.90 | 45.52 | 27.30 | 27.66 | 58.52 | 16.05 |
| LargeSet | $RTrain_p$ | 85.94 | 16.41 | 75.35 | 85.94 | 16.41 | 75.35 | 80.16 | 19.50 | 66.89 | 62.74 | 29.28 | 45.71 | 50.96 | 34.90 | 34.19 |
| | InfoNet | **86.66** | **15.77** | **76.46** | **86.66** | **15.77** | **76.46** | **80.82** | **19.55** | **67.81** | **64.28** | **28.62** | **47.36** | **53.61** | **33.12** | **36.62** |

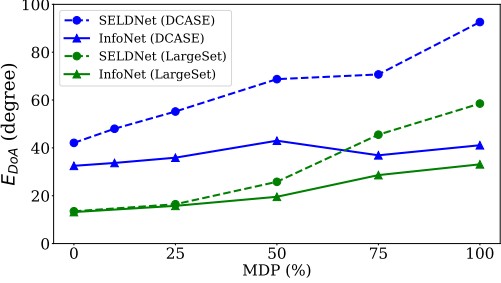

Figure 4: Effect of MDP on the $E_{DoA}$ for the DCASE and LargeSet.

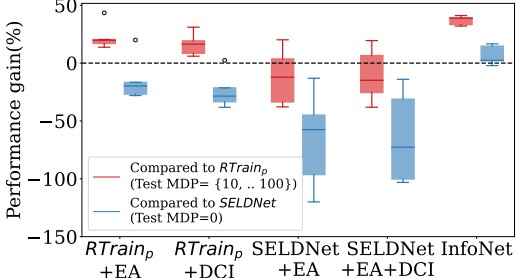

Figure 5: Improvement of $E_{DoA}$ on DCASE for different model configurations with Test MDP = $\{10, 25, 50, 75, 100\}$%.

Table 1 and Figure 4 compares the $E_{DoA}$, $F1_{20}$, and $R_L$ of **InfoNet**, $RTrain_p$ and SELDNet for a set of MDP with $p = 25, 50, 75, 100$% on DCASE and LargeSet. We observe that **InfoNet** achieves $38.48 \pm 11.86$% and $22.55 \pm 18.30$% degree lower $E_{DoA}$ than *SELDNet*

on the DCASE and LargeSet, respectively. We can also find a consistent $F1_{20}$ and $R_L$ form table 1, which signifies the **InfoNet**'s ability to identify all data points in a relevant class consistently.. Figure 4 shows the performance trend for both datasets on the presence of corrupted input streams; hence lower information. It is evident that, with the higher percentage of missing information, the *SELDNet* tends to suffer while **InfoNet** performs fairly consistently by retrieving the missing information by maintaining a 4.39 and 7.89 $E_{DoA}$ variance for DCASE and LargeSet, respectively.

In Table 1, Sys achieves $33.24 \pm 6.66\%$ and $2.98 \pm 2.09\%$ degree lower $E_{DoA}$ than $RTrain_p$ on the DCASE and LargeSet, respectively. Similar to the previous dataset the consistency of $F1_{20}$ and $R_L$ are evident. The $E_{DoA}$ of both and $RTrain_p$ SELDNet significantly increases with lower available information or higher $p$ since spatial relationships among different input streams are the most critical feature for localization tasks. Table 1 shows that the improvement is lesser for LargeSet than the DCASE dataset. This can be attributed to the fact that, as LargeSet doesn't have overlapping sound sources, it has a lesser effect of MDP also; hence, there is less scope for improvement.

### 5.3 ABLATION STUDY

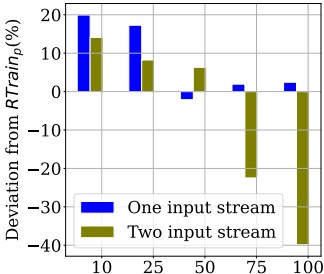

Figure 6: Effect of simultaneously missing mics.

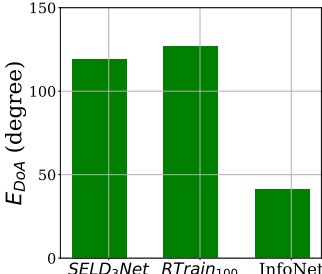

Figure 7: Effect of less number of mic.

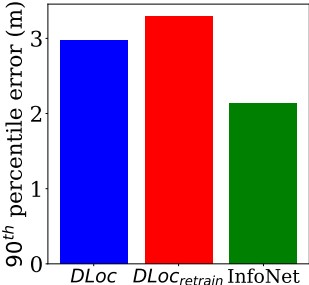

Figure 8: Result of WiFi localization.

**Effect of EA.** To understand the impact of the EA only, we integrate EA with the downstream task and jointly train them with p% missing information($RTrain_p + EA$). We observe that the $E_{DoA}$ reduces by $22.85 \pm 10.56\%$ than only in $RTrain_p$ (Red box in Figure 5). Though adding EA improves the performance, it can only estimate the available information entropy at each element of the feature matrix, $\tilde{F}$. The failure to estimate the missing elements' value in the full-rank version, $F$, lowers the $E_{DoA}$ by $14.25 \pm 17.62\%$ than original SELDNet with no missing information as input (Blue box in Figure 5). Thus, **InfoNet** with EA and DCI estimates the feature values have $16.76 \pm 11.21\%$ less $E_{DoA}$ than $RTrain_p + EA$.

**Effect of DCI.** Figure 5 shows only adding DCI with $RTrain_p$ ($RTrain_p + $ DCI) reduces $E_{DoA}$ by $16.22 \pm 8.87\%$ compared to only $RTrain_p$ (red box)., However, only $RTrain_p + $ DCI performs worse when no information was missing and has $26.55 \pm 14.56\%$ higher $E_{DoA}$ than *SELDNet* because DCI estimates the full-rank features ($\bar{\bar{F}}$), it adds error to the matrix elements where the effect of missing data is minimal. EA provides the information entropy and reduces this additional error. **InfoNet** replaces the low-rank feature with the guidance of estimated information entropy.

**Effect of Re-Training Downstream Task.** To understand the benefit of retraining the downstream task, we only add a trained EA with pre-trained *SELDNet* whose weights are frozen ($SELDNet + EA$). Figure 5 shows that $SELDNet + EA$ significantly increases $E_{DoA}$ by $54.41 \pm 49.79\%$ than $RTrain_p$. Adding both EA and DCI with pre-trained *SELDNet* ($SELDNet + EA + DCI$) reduces the $E_{DoA}$ by $53.76 \pm 46.52\%$ than $RTrain_p$. Thus, jointly training EA and downstream task is necessary for EA to learn the element-wise information entropy, which is critical for successfully replacing low-rank features.

**Effect of Simultaneously Missing Input Streams or Mics.** Figure 6 compares two systems with one and two simultaneously missing mics or input streams. As anticipated, single missing mic system performs better than the two missing mics system due to the

lack of available information for estimation. Though missing two mics results in 46.18% less $E_{DoA}$ than a single missing mic for 50% MDP, with higher MDP, the $E_{DoA}$ increases significantly because with more simultaneously unavailable mics, the information entropy estimation becomes more inexact, and the guided replacement becomes inaccurate.

**Effect of Less Number of Input Streams or Mics.** Figure 7 observes the effect of having one less mic or input stream. When we have full-rank input from only three mics in training and testing ($SELD_3Net$), the $E_{DoA}$ 6.29% lower than $RTrain_{100}$, where the information missing happening at various mics. **InfoNet** reduces the $E_{DoA}$ by 67.59% from $SELD_3Net$ and achieves 22.80% $E_{DoA}$ gain than SELDNet at 0% MDP.

## 5.4 Performance on WiFi CSI.

To verify the effectiveness of the proposed **InfoNet** in different data domains, we evaluate it on a state-of-the-art large-scale location-referenced WiFi CSI dataset Ayyalasomayajula et al. (2020) for user localization with 75% of missing information. This dataset consists of two indoor environments spanning 2000 sq. ft. area under 8 different scenarios. We adhere to the same two-step image-based representation feature extraction process proposed by the authors along with their proposed a deep learning-based wireless localization algorithm (DLoc), which is an encoder-decoder architecture with two parallel decoders. Detailed architecture of DLoc is provided ar appendix C.2. Figure 8 shows that the baseline $DLoc$'s $90^{th}$ percentile error is 2.98m. Though retraining $DLoc$ ($DLoc_{retrain}$ increases it to 3.29m, **InfoNet** reduces it to 2.13m, which signifies 35.25% and 25.52% of performance gain over $DLoc_{retrain}$ and $DLoc$.

## 6 Related Work

The mutual information between the input and output of each layer quantifies DNN Colombini et al. (2014). This information is hard to understand as we only have the data samples but not the distributions McAllester & Stratos (2020). Previously proposed optimizers solve the tasks with minimal knowledge of input distribution Alemi et al. (2016) but do not address corrupted data samples.

Attention learns the cognitive and behavioral characteristics to focus on essential information and ignore the rest selectively Colombini et al. (2014). For multi-channel input, attention explores the channel characteristics and estimates the channel state information Gao et al. (2021). Though commonly used later in a network, early attention improves performance with substantial margins Hajavi & Etemad (2020) motivating our EA component.

Corrupted multi-stream data is used to attain better performance on tasks such as speech enhancement Taherian et al. (2022). Unlike source localization, speech enhancement only partially depends on hard-to-recover spatial information. Some recent works focus on recovering missing modalities in multimodal sensing Ma et al. (2022; 2021).

Classic sound source localization (SSL) methods, e.g., MUSIC Gupta & Kar (2015), independent component analysis Noohi et al. (2013), and sparse models Yang et al. (2018), perform poorly for under-determined scenarios. Recently DNN has shown promising results for SSL Ferguson et al. (2018); Yiwere & Rhee (2017); Hirvonen (2015). However, none of the existing sound source localization works focuses on missing or corrupted information.

## 7 Discussions and Limitations

Interdependence among different random variables can be utilized to recover missing information. Classical algorithms to handle missing data points are insufficient due to their limitations, e.g., the requirement of prior knowledge and performance degradation on high and complex data distributions. This paper shows how to overcome such limitations. However, the proposed method requires training the downstream task to gain performance. Also, we have minimal improvement for the task where multiple data stream is optional. We will focus on removing the downstream task retraining requirement in our future work.

## Reproducibility Statement

We conduct this research study by keeping reproducibility in mind. In this section, we describe several aspects of implementation and datasets used in the experiments.

**Codebase.** The implementation of **InfoNet** with the LargeSet dataset can be found here (`https://github.com/hidethyself/InfoNet`). Our feature extraction pipeline is motivated from Adavanne et al. (2018). Also, we follow the feature extraction procedure and SELDNet implementation from here (`https://github.com/sharathadavanne/seld-dcase2022`).

For WIFI-CSI dataset, we follow the implementation provided by the authors (`https://github.com/ucsdwcsng/DLoc_pt_code`) to implement the Wloc baseline. Though the feature extraction procedure is not available in this codebase, the authors graciously provided us with the code for feature extraction from raw data upon request.

**Dataset.** We use three datasets in this research.

- **DCASE2021 - Task 4**: This dataset is available at (`https://dcase.community/challenge2021/task-sound-event-localization-and-detection`)
- **LargeSet**: We simulated 10 different environments to create a 50 hours of audio localization dataset. The detailed procedure of this simulation can be found here A. The dataset is also available at (`https://drive.google.com/drive/folders/1YbOdBA8p-WI_FRT7ktbiex3TXhZS7igb?usp=drive_link`)
- **WIFI-CSI**: The dataset is available here (`https://wcsng.ucsd.edu/dloc/`)

**Parameters & Environment.** The hyper-parameter used in the experiment (for feature extraction & model) is stated in the provided codebase. We also provide the required package list for this project in the codebase. The data split (train, validation, and test) is only done once to train and test the model with the same set of instances for all the MDPs. We also provide these split lists in the provided dataset.

**Hardware.** We conducted all the experiments on a machine with the following configuration.
**Operating Ststem:** Ubuntu 20.04
**Processor:** Ryzen Threadripper 3960X - 24-Core - 3.8 GHz / 4.5GHz Boost - 280w.
**RAM:** 128 GB DDR4 3200MHz
**GPU:** 2x NVIDIA RTX 3090, 10496 CUDA Cores, 24GB GDDR6X Memory, PCIe 4.0

**Instruction.** Detailed instruction to run the experiments is provided in the codebase.

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

## A  Data Generation Details for LargeSet

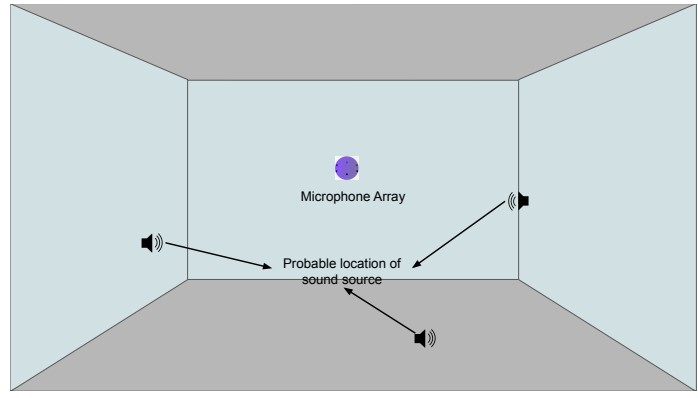

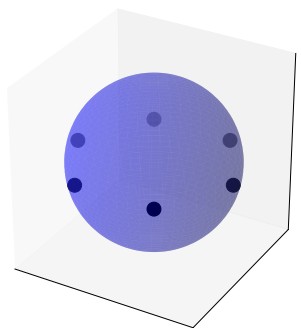

Figure 9: Simulation environment with the microphone array in the center.

Figure 10: Microphone array

We use the configurations in table 2 to simulate 10 different environments. Each environment consists of a room with a ceiling, floor, and wall with different elements. Each room is box-shaped with $(l \times w \times h)$ m dimensions.

A microphone array of 6 microphone is placed at the $(\frac{l}{2}, \frac{w}{2}, \frac{h}{2})$ co-ordinate of the room, which is the center the room. The microphone is a sphere in shape. The microphones are equally spaced on the circumference of the sphere. The radius of the sphere is 4.25cm. Figure 9 and 10 show the simulation environment and microphone array structure with 6 microphones.

We use *UrbanSound8K* Salamon et al. (2014) dataset as the different sound source present in the simulation environments. We randomly generate a point $(x, y, z)$ inside the room as the originating point of the sound. This generated point is used as the ground truth for training the model. Here,

$$x \in [0, l)$$
$$y \in [0, w)$$
$$z \in [0, h)$$

We use *PyRoomacoustics* Scheibler et al. (2018) to simulate the different scenarios. The audios are sampled at 44100 Hz. For each environment, we simulate 5 hours of audio; hence, a total of 50 hours of audio was recorded for 10 environment.

Table 2: Configurations of all the simulated environments.

| # | Room Dimension($lm \times wm \times hm$) | Ceiling | Floor | Wall |
|---|---|---|---|---|
| 1 | $10 \times 7.5 \times 3.5$ | unpainted_concrete | carpet_6mm_open_cell_foam | reverb_chamber |
| 2 | $10 \times 7.5 \times 3.5$ | plasterboard | cocos_fibre_roll_29mm | ceramic_tiles |
| 3 | $15 \times 8 \times 5$ | wooden_lining | linoleum_on_concrete | rough_lime_wash |
| 4 | $15 \times 8 \times 5$ | hard_surface | carpet_rubber_5mm | brickwork |
| 5 | $10 \times 7.5 \times 3.5$ | plasterboard | carpet_hairy | wooden_lining |
| 6 | $10 \times 7.5 \times 3.5$ | rough_lime_wash | carpet_6mm_closed_cell_foam | plasterboard |
| 7 | $15 \times 8 \times 5$ | rough_lime_wash | felt_5mm | brick_wall_rough |
| 8 | $15 \times 8 \times 5$ | rough_lime_wash | carpet_cotton | unpainted_concrete |
| 9 | $7.5 \times 4 \times 4.3$ | plasterboard | carpet_thin | rough_concrete |
| 10 | $7.5 \times 4 \times 4.3$ | unpainted_concrete | carpet_tufted_9.5mm | hard_surface |

# B EVALUATION METRICS

## B.1 DEGREE OF ARRIVAL ESTIMATION ERROR ($E_{DoA}$)

$E_{DoA}$ of entire dataset is represented by

$$E_{DoA} = \frac{1}{D} \sum_{d=1}^{D} \zeta((x_e^d, y_e^d, z_e^d), (x_r^d, y_r^d, z_r^d))$$

Here, $(x_e, y_e, z_e)$ and $(x_r, y_r, z_r)$ are the estimated and reference coordinates, respectively. $\zeta$ is the angle between the reference and estimated coordinates. Now, $\Delta x = x_r - x_e$, $\Delta y = y_r - y_e$, $\Delta z = z_r - z_e$. $\zeta$ calculates the angle between $d^{th}$ estimated and reference DoAs. $\zeta$ can be written as –

$$\zeta = 2 \arcsin\left(\frac{\sqrt{\Delta x^2 + \Delta y^2 + \Delta z^2}}{2}\right) \frac{180}{\pi}$$

## B.2 EVENT DETECTION F1-SCORE ($F1$)

$F1$ is defined as follows

$$F1 = \frac{2 \sum_{k=1}^{K} TP(k)}{2 \sum_{k=1}^{K} TP(k) + \sum_{k=1}^{K} FP(k) + \sum_{k=1}^{K} FN(k)}$$

Here, for the $k^{th}$ one-second segment, $TP(k)$ is the total number of sound events present in both prediction and reference, $FP(k)$ represents the number of active sound events in prediction, but inactive in reference, $FN(k)$ stands for the number of sound events inactive in the prediction.

## B.3 LOCALIZATION RECALL ($R_L$)

$R_L$ measures the localization performance on each frame. $R_L$ is represented by

$$R_L = \frac{TP_{DoA}}{TP_{DoA} + FN_{DoA}}$$

Here, $TP_{DoA}$ stands for the total number of time frames where number of estimated DoAs is equal to the number of reference DoAs, and $FN_{DoA}$ represents the total number of time frames where estimated and reference DoAs are not equal. We consider DoA prediction as a true positive if its distance is within $20°$.

## C    BASELINE MODEL ARCHITECTURE

### C.1    SOUND EVENT LOCALIZATION & DETECTION NETWORK (SELDNET)

We employ the SELD algorithm from Adavanne et al. (2018); Shimada et al. (2021) to develop our downstream task model. This model uses an activity-coupled Cartesian DOA (ACCDOA) representation, which assigns a sound event activity to the length of a corresponding Cartesian DOA vector. The SELD task is converted into an ACCDOA estimation problem. The model comprises a convolutional recurrent neural network (CRNN) Cao et al. (2019) followed by a single full-connected (FC) layer, which estimates the ACCDOA representation vector.

The feature sequence of $T$ frames, with an overall dimension of $T \times M/2 \times 2C$, where the 2C dimension consists of C magnitude and C phase components. Here, the $M/2$ positive frequencies without the zeroth bin are used.

SELDNet uses local shift-invariant features from the spectrograms. To learn these features, SELDNet uses multiple layers of CNN layers. Each CNN layer consists of $P$ filters of $3 \times 3 \times 2C$ dimension. These filters work in the time and frequency axis. Each CNN layer is followed by a rectified linear unit (ReLU) activation and batch normalization layer. To reduce the dimensionality, max-pooling is applied. But max-pooling is only applied along the frequency axis to keep the sequence length $T$. The output shape of the CNN blocks is $T \times 2 \times P$. This CNN output is further reshaped to a $T$ frame of length sequence $2P$ to feed into the bidirectional RNN layer. With the introduction of the RNN layer, the model learns the temporal features. A gated recurrent unit (GRU) is used here as the RNN layer. Finally, a fully connected layer consists of $3N$ nodes with tanh activation, where each of the $N$ sound event classes is represented by 3 nodes corresponding to the sound event location in $x$, $y$, and $z$, respectively.

### C.2    DEEP LEARNING BASED WIRELESS LOCALIZATION (DLOC)

DLoc Ayyalasomayajula et al. (2020) follows a single encoder and two decoder architecture. One of the decoders is named a consistency decoder, and the other is a location decoder. The encoder takes the input heatmaps of all corresponding APs and generates a concise representation. This concise representation then goes into two decoders. The consistency decoder ensures that the network sees a consistent view of the environment across different training samples and access points. The location decoder takes the encoder representation and estimates a user's location.

ResNet blocks are the main building blocks of encoders and decoders. The encoder consists of 6 ResNet blocks. It also has an initial convolution layer with $7 \times 7$ kernel followed by tanh activation. The consistency and location decoders have 6 and 3 ResNet blocks, respectively. For the inference, only the location decoder is used.

# D EXTENDED RESULTS

## D.1 EFFECT OF DIFFERENT MODEL CONFIGURATION AT VARIOUS MDP

Table 3: Performance with different model configurations on DCASE dataset with $p\%$ MDP.

| MDP | 10 | | | 25 | | | 50 | | | 75 | | | 100 | | |
|---|---|---|---|---|---|---|---|---|---|---|---|---|---|---|---|
| Model Configuration | $F1_{20}$ | $E_{DoA}$ | $R_L$ | $F1_{20}$ | $E_{DoA}$ | $R_L$ | $F1_{20}$ | $E_{DoA}$ | $R_L$ | $F1_{20}$ | $E_{DoA}$ | $R_L$ | $F1_{20}$ | $E_{DoA}$ | $R_L$ |
| $RTrain_p$ + EA | 16.20 | 33.70 | 38.50 | 13.70 | 49.10 | 35.00 | 12.50 | 50.40 | 30.80 | 11.30 | 53.40 | 31.50 | 11.20 | 53.90 | 32.10 |
| $RTrain_p$ + DCI | 9.50 | 41.10 | 28.10 | 6.60 | 54.10 | 24.30 | 6.40 | 51.10 | 23.70 | 5.20 | 58.20 | 24.10 | 6.90 | 56.20 | 26.50 |
| SELDNet + EA + DCI | 10.10 | 47.60 | 26.60 | 7.70 | 66.30 | 22.40 | 4.30 | 61.00 | 15.80 | 2.40 | 82.70 | 12.80 | 1.50 | 92.60 | 13.50 |
| SELDNet + EA | 10.90 | 48.00 | 27.00 | 8.20 | 55.20 | 21.70 | 4.70 | 72.70 | 15.30 | 1.70 | 85.50 | 11.70 | 2.60 | 84.30 | 11.40 |

Table 3 shows the effect of different (EA, DCI) on the result. The result shows only EA and DCI can not retrieve the missing information. GR combines them, and performance is improved.

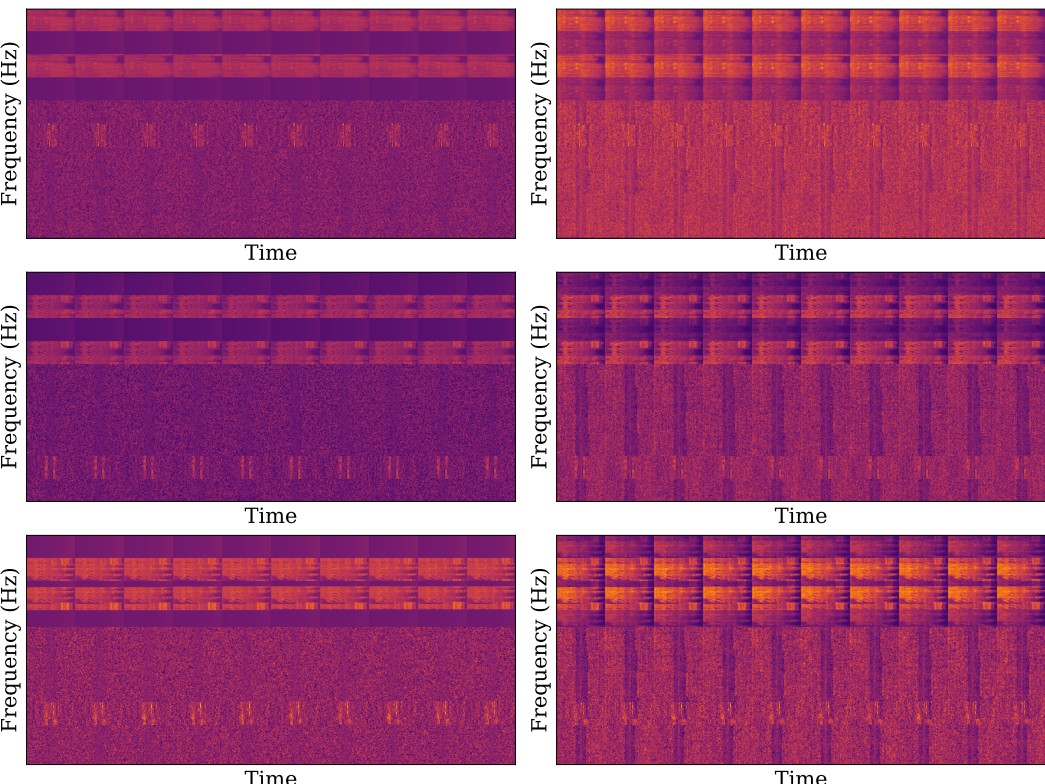

Figure 11: Example of low-rank (left) and estimated full-rank (right) features from the DCASE test set.

## D.2 VISUAL REPRESENTATION OF GUIDED REPLACEMENT OF **InfoNet**

Figure 11 shows a few examples of the low-rank $\tilde{F}$ and estimated full-rank features $\hat{F}$. **InfoNet** receives available information estimation $I$ from EA and interpolated feature $\bar{\bar{F}}$ from DCI. Then, the guided replacement (GR) block replaces unavailable information according to the equation 7 by using $I$ , $\bar{\bar{F}}$ and $\tilde{F}$. From the figure, it is evident that how **InfoNet** successfully recovers the missing information achieves better performance.

