# OpenReview forum: "InfoNet: Missing Information Retrieval in Multi-Stream Sensing Systems"
_ICLR.cc/2024/Conference — ICLR 2024 Conference Withdrawn Submission_

### Official Review · Reviewer_fMm6 · 2023-10-29

**Soundness:** 2 fair
**Presentation:** 2 fair
**Contribution:** 1 poor
**Rating:** 1
**Confidence:** 4

**Summary:**

The authors propose a network architecture to try to recover time-series data for multi-stream tasks such as multi-microphone audio processing. The authors give some theoretical insights on the reasons behind suggesting the proposed architecture which resembles a data interpolation scheme conditioned on the per-sample information entropy. The proposed method shows that the usage of the extra modules that estimate the information entropy and the one that tries to perform interpolation between data streams performs better than a model without those modules for the tasks which are considered (e.g. multi-microphone estimation of arrival and multi-microphone sound event detection).

**Strengths:**

I like the principled way that the authors tried to introduce their model by using a module that specifically tries to estimate the uncertainty of one value and another module that can use the previous results to generate the missing data (data streams with high uncertainty) using conditional interpolation.

**Weaknesses:**

- First of all, the early attention framework which performs the explicit estimation of the uncertainty of the data-streams was not compared to other alternative modules that could perform implicit uncertainty estimation of the data [A] and feature recovery (e.g. any GAN, VAE, diffusion generative model or even a simple discriminatively trained model using simple regression on the missing data-streams).

- The authors completely omit to compare with even a single decent benchmark model in the tasks of degree of arrival estimation. There are hundreds of models that the authors could use to just show how their models compare against them, I will just cite some [B, C, D, E]. There is no way of knowing that this whole implicit uncertainty estimation and conditional interpolation would yield a higher performance over a simple baseline. The authors should explicitly describe how they applied the baseline models of the literature to this task and use widely known benchmark datasets to compare against.

- The paper becomes almost unreadable with the extreme overuse of acronyms throughout the manuscript. The pages are more than enough to present everything without reducing the readability of the paper.

I would be more than happy to increase my score if all the above weaknesses are addressed by the authors.

[A] Ma, C., Li, Y. and Hernández-Lobato, J.M., 2019, May. Variational implicit processes. In International Conference on Machine Learning (pp. 4222-4233). PMLR.

[B] Sharath Adavanne, Archontis Politis and Tuomas Virtanen, 'Direction of arrival estimation for multiple sound sources using convolutional recurrent neural network' at European Signal Processing Conference (EUSIPCO 2018)

[C] Sharath Adavanne, Archontis Politis and Tuomas Virtanen, 'Multichannel sound event detection using 3D convolutional neural networks for learning inter-channel features' at International Joint Conference on Neural Networks (IJCNN 2018)

[D] Serizel, R., Turpault, N., Shah, A. and Salamon, J., 2020, May. Sound event detection in synthetic domestic environments. In ICASSP 2020-2020 IEEE International Conference on Acoustics, Speech and Signal Processing (ICASSP) (pp. 86-90). IEEE.

[E] Papageorgiou, G.K., Sellathurai, M. and Eldar, Y.C., 2021. Deep networks for direction-of-arrival estimation in low SNR. IEEE Transactions on Signal Processing, 69, pp.3714-3729.

**Questions:**

1. What is the purpose of Figure 11? The authors mention that: "From the figure, it is evident that how InfoNet
successfully recovers the missing information achieves better performance.” How did the authors arrive at this conclusion? Better performance compared to some other model or some other method?

---

### Official Review · Reviewer_tZQw · 2023-11-01

**Soundness:** 3 good
**Presentation:** 2 fair
**Contribution:** 2 fair
**Rating:** 3
**Confidence:** 3

**Summary:**

This paper proposes InfoNet, a generalized algorithm that retrieves the information from a corrupted feature set to recover the inference performance loss due to corrupted input data streams. InfoNet estimates the information entropy at every element of the input feature to the network and retrieves the missing information in the input feature matrix. InfoNet is tested for two downstream applications, including sound source localization and wireless signal-based source localization, and has shown superior performance over baselines in corrupted data schemes.

**Strengths:**

Missing or corrupted information under multi-stream sensors is a common setting in real life and has not been studied in sound source localization. The proposed method, based on estimation of information entropy is intuitive and has outperformed several baselines under this setting. The paper also conducts a thorough analysis of the proposed model.

**Weaknesses:**

1. Lack of stronger baselines. The proposed method is compared only to two simple baselines - retraining with missing streams and a basic time-domain recovery method. More recent approaches, such as those in [1] and [2], are not included in the comparison.
2. The model is only tested under limited data setting (<100 hours).  It's unclear how the model performs in larger datasets, even though practical use cases often involve much more data (e.g., Audioset contains ~5k hours of multimodal signals)
3. The downstream model for sound event localization is quite basic. Its performance with more recent sound event localizers (e.g., [3][4]) is unclear.


[1]. Ma et al., SMIL: Multimodal Learning with Severely Missing Modality, in AAAI 2021

[2]. Wang et al., Multi-modal Learning with Missing Modality via Shared-Specific Feature Modelling, in CVPR 2023

[3]. Shimada,  et al., ACCDOA: Activity-coupled cartesian direction of arrival representation for sound event localization and detection, in ICASSP 2021

[4]. Cao et al., An improved event-independent network for polyphonic sound event localization and detection, in ICASSP 2021

**Questions:**

1. How does the model compare to other methods, such as those in [1] and [2], in sound event localization or other multimodal tasks like Audiovision-MNIST?
2. How does the model perform in a larger data setting (e.g., audio-visual event classification)?

---

### Official Review · Reviewer_9qjF · 2023-11-02

**Soundness:** 2 fair
**Presentation:** 3 good
**Contribution:** 2 fair
**Rating:** 5
**Confidence:** 4

**Summary:**

* This paper presents an innovative algorithm, InfoNet, which addresses the challenge of corrupted sensor data in Deep Neural Networks (DNNs). InfoNet's novelty lies in its components: Early Attention (EA) for pinpointing problematic data, Deep Conditional Interpolator (DCI) for reconstructing missing information, and Guided Replacement (GR) for restoring data integrity. These innovations enable InfoNet to estimate information entropy and effectively retrieve missing data, thereby aiding in the recovery of DNN performance.
* The algorithm's effectiveness is validated through sound localization tests, with its potential applicability to other sensing tasks, such as wireless localization, also being explored.
* The paper details these novel components, the experimental setup, and discusses the broader implications for advanced techniques in managing incomplete or corrupted data inputs for DNN applications.

**Strengths:**

-	Novel Components: InfoNet's innovative approach is underscored by its unique components—Early Attention (EA) for targeted data analysis, Deep Conditional Interpolator (DCI) for accurate data reconstruction, and Guided Replacement (GR) for precise data input restoration. These improvements collectively enhance the algorithm's capability to estimate information entropy and retrieve missing information in multi-stream sensing systems, potentially addressing the challenges of corrupted data inputs in deep learning.
-	Generalizability: The algorithm's utility extends beyond audio streams, as demonstrated by its potential for application in other sensing modalities, such as wireless signal-based source localization.
-	Detailed Evaluation: Extensive experimental evaluations provide a comprehensive comparison of the InfoNet algorithm against baseline methods. The results, highlighting significant performance improvements specifically in reducing localization errors, attest to the algorithm's effectiveness.
-	Well Written: The paper is well-structured with a clear writing style.
-	Soundness: The theoretical approach of the paper seems sound, and the method is well-grounded in information theory, specifically leveraging the concept of entropy to address missing information in multi-stream sensing systems.

**Weaknesses:**

-	The paper's experimental section adequately covers comparisons with baseline methods, but primarily takes SELDNet[1] from 2018 for its most recent comparison. This raises a question about whether the paper might have missed evaluating InfoNet against the latest developments in sound event localization and detection, which would provide a clearer picture of its relative performance to latest methodologies.
-	The current approach, which combines concepts from information theory and feature extraction methods similar to U-Net, may not sufficiently demonstrate originality due to its simplicity. Incorporating more unique or novel elements into the proposed method could help enhance its originality and distinguish it from existing approaches.
-	While the experiments performed within the paper align with its scope, the absence of experiments involving other commonly encountered modalities such as images, videos, and text restricts the demonstration of the method's general applicability. If experiments in these additional modalities could be included, the authors can showcase the versatility and effectiveness of their approach across a broader range of data types.

[1] Adavanne, Sharath, et al. "Sound event localization and detection of overlapping sources using convolutional recurrent neural networks." IEEE Journal of Selected Topics in Signal Processing 13.1 (2018): 34-48.

**Questions:**

* The method appears to be a straightforward application of entropy concepts from information theory, combined with a U-Net-like feature extraction process. There's a concern about the overall originality and innovation of the approach.
* Limited Experimentation and Potential Overstatement in Title: Experiments are primarily focused on sound localization with corrupted audio streams or wireless signal-based source localization. The absence of experiments in other modalities (like images, videos, and text) limits the understanding of the method's applicability and effectiveness across different domains.
* The paper doesn't clearly specify what aspects are “highlighted” in Figure 1. This lack of clarity makes it difficult to understand the specific innovative elements or key features being presented.
* Baseline's model structure is also quite simple; I question if it is because there is no up to date research in this direction.
* Is there anything important about the “stream” in the title? It doesn't seem to require a lot of real-time?
* What is the computational cost of InfoNet, especially when dealing with real-time data streams? Is the increase in performance justified by the computational overhead?